# Causes of the Shortage of Physics Teachers in Croatia

**Nataša Erceg** [1,*] , **Lejla Jelovica** [2,3] , **Vanes Mešić** [4] , **Ljubiša Nešić** [5] , **Ivana Poljančić Beljan** [1]
and **Patricija Nikolaus** [3]

1. Faculty of Physics, University of Rijeka, 51000 Rijeka, Croatia; ipoljancic@phy.uniri.hr
2. Faculty of Health Studies, University of Rijeka, 51000 Rijeka, Croatia; lejla.jelovica@uniri.hr
3. Faculty of Science, University of Split, 21000 Split, Croatia; pnikolaus@pmfst.hr
4. Faculty of Science, University of Sarajevo, 71000 Sarajevo, Bosnia and Herzegovina; vanes.mesic@gmail.com
5. Faculty of Sciences and Mathematics, University of Niš, 18000 Niš, Serbia; ljubisa.nesic66@gmail.com
* Correspondence: nerceg@phy.uniri.hr

**Abstract:** Knowing the causes of the shortage of physics teachers in primary and secondary schools is necessary for the development of effective educational policies because the shortage of physics teachers is a global and persistent problem with negative consequences for the quality of education, but also for the survival of the physics profession as a whole. The aim of this study was to investigate, for the first time, the opinions of Croatian physics teachers on the causes of the deficit in their profession. For this purpose, we conducted a descriptive cross-sectional study using an online survey with Likert-type items and a constructed response item. A total of 390 respondents from all over Croatia participated in the survey, which is 29% of the total population of Croatian physics teachers in the 2022/2023 school year. According to their opinion, the causes of the shortage of Croatian physics teachers are related to the following: lack of incentives and support from the relevant institutions and bodies, the inadequacy of physics content in the curricula, the lack of motivation and negative attitude of students towards physics, impeded professional development, unequal opportunities, and challenges such as excessive workload. The results obtained provide a concrete basis for the development of an effective policy to solve the problem of the physics teacher shortage in Croatia and beyond by governments, universities, and schools.

**Keywords:** physics education; causes of physics teacher shortage; physics teachers; elementary (lower secondary) school; secondary (upper secondary) school

## 1. Introduction

The shortage of physics teachers is a global and persistent problem that has been attempted to be solved in different ways in different countries based on the results of studies conducted by relevant institutions (e.g., [1]). In science, technology, engineering, and mathematics (STEM), there is a significant shortage of teachers (e.g., [2]), with the shortage of physics teachers being one of the most severe.

Identifying the factors that affect the shortage of professional physics teachers [3] may help the educational stakeholders to develop more effective policies related to attracting and retaining competent physics teacher candidates. Thereby, a useful first step would be to identify factors that influence students' career choices.

Hillier et al. [4] conducted a study on the career choices of respondents who had graduated as physics teachers in the United Kingdom (UK). Two main factors were found to have a positive influence on recruitment and retention in the physics teaching profession. The first important factor was the initial motivation to enter the teaching profession, often due to personal positive teaching experiences [2,5]. Another significant factor was integration into the physics teaching profession with the help of another experienced physics teacher. Other reasons for entering and staying in the physics teaching profession were also discovered, such as the teaching tradition in the family, the attractive perception

of teaching as a profession that is adapted to family life [6–8], and the love of working with children [6–12]. The advantages of the teaching profession compared to other professions also include a shorter duration of the working day and a longer duration of vacations [12].

Personality type can also influence the choice of study and career [13,14]. For example, people who choose to study physics tend to be introverts [15], so they are unlikely to find teaching children enjoyable [16]. Therefore, they choose teaching as a profession to a lesser extent, which is in line with the findings by Ellis [17].

One of the reasons for the shortage of physics teachers could be also the gender imbalance in choosing a teaching profession. For example, the majority of individuals who choose to become teachers are women [18,19] and the majority of individuals who choose to study physics in the UK are men [20].

The main factors that negatively affect recruitment and retention of physics teachers include low salaries and incentives [1,18,19,21] that do not cover the cost of living [19,22,23], and the so-called single salary schedule [12]. Single salary schedule means that all teachers with the same level of education and professional experience receive the same salary, regardless of the subject they teach. The situation is different in the business world, where physics graduates can find much more lucrative jobs compared to working in schools [19]. Indeed, physicists (as well as chemists and mathematicians) have more opportunities and higher salaries compared to other professions [3,12]. One of the reasons for this is that only a relatively small number of people enroll in science and mathematics studies, which require a rare kind of talent and a high level of effort [12,24]. For example, physics requires a strong mathematical background and contains more abstract concepts than many other fields [3]. This is not to say that prominent professionals in various fields do not expend much talent and effort during their studies. However, hardly anyone will deny that an undergraduate degree in mathematics, physics, or chemistry is more demanding and rigorous from the perspective of an average student than, for example, a degree in the humanities or social sciences [12].

Additional negative influences on physicists' employment and retention in the teaching profession include lack of professional development opportunities [3], high workload [3,6,19,22,23], low morale, perception of low status, low self-esteem, poor public image, undisciplined students, poor working conditions [19,22,23], and demographics, i.e., changes in the school-age population and the age group from which teachers are recruited [19].

Universities are also mentioned as being responsible for the shortage of physics teachers because they do not have enough degree programs to prepare future physics teachers [25] and do not produce enough graduate physics teachers [26]. However, the recruitment of a greater number of science and mathematics teachers does not change the fact that they often pursue careers outside of school, after graduating [12]. As long as salaries offered by employers in other sectors of the labor market are significantly higher than those offered to teachers, a relatively small number of science and math graduates will choose to become or remain teachers. Even though there are incentives available, such as government loans for individuals aspiring to become science and mathematics teachers, as well as funds allocated for retraining teachers from different fields to specialize in science and mathematics teaching, the allure of higher earnings in alternative professions makes these options comparatively less appealing. In addition, to attract the workforce, some employers offer to pay off the loans of potential employees [12].

Another issue is that due to the modest salary level of teachers, there is a tendency to enroll students with low skills in teacher study programs [27–29]. Taken together with the low quality of university curricula [3], this results in failing to produce quality personnel to meet the growing demands in schools [26]. Such inadequate preparation of physics teachers may, for example, negatively affect (a) students' scores on international tests in physics [30–32], (b) enrollment in physics and STEM courses [33], and (c) the development of physics, facing the STEM workforce with the challenges and demands of the 21st century [34].

Although numerous causes of physics teacher shortages have been identified, they do not apply to every geographic area; therefore, they cannot be generalized [35]. For example, in comparing the situation in different regions of the UK where there is a different need for physics teachers, Sparkes found that the previously mentioned causes (e.g., demographics, cost of living, ability to employ physicists in other professions with higher salaries, gender, low morale, demanding studies) sometimes lead to a shortage of physics teachers and sometimes do not. This observation partly explains the unresolved and widespread nature of this problem worldwide [36], because the good practices of certain countries cannot simply be transferred to other countries.

In Croatia, there is also the problem of a shortage of qualified physics teachers, which not only negatively affects the students' learning of physics, but also questions the survival of the entire profession of physicists [37]. For decades, the educational stakeholders in Croatia attempted to solve this problem by relying on a trial-and-error approach instead of relying on relevant educational research. Motivated by this fact, we conducted a descriptive cross-sectional study [38] that aimed to investigate the opinions of Croatian physics teachers on the causes of the deficit in their profession. This is a potentially important contribution to physics education literature because it is the first study on the causes of the physics teacher shortage within the Croatian context. Taken together with the findings from earlier research on the characteristics of physics teachers in Croatia [37], the results from this research provide a valuable starting point to solving the problem of the physics teacher shortage in Croatia.

## 2. Materials and Methods

To fulfill the above-mentioned research aim, we conducted a descriptive cross-sectional study [38]. We conducted an online survey with Likert-type items and a constructed response item. Based on teachers' answers to these items, we could check to what extent the causes of physics teacher shortage that had been identified in international research are relevant for the Croatian context. In addition, the answers on the constructed response item helped us to gain deeper insights on potential causes of physics teacher shortage that are specific for the Croatian context. Consequently, our study may be considered as mixed research with an emphasis on the quantitative approach [39,40]. In fact, we decided to use a large teacher sample and a preselected assessment instrument, as well as to heavily rely on the statistical analyses of numeric data. However, in order to further elaborate, illustrate, enhance, and clarify the findings from our research, we included direct quotations from research participants (i.e., their answers to the open-ended question) in our paper, which is a prominent characteristic of qualitative research.

### 2.1. Sample

A total of 390 respondents participated in the survey, representing 29.13% of the total population of Croatian physics teachers (*N* = 1339) in the 2022/2023 school year.

The total number of physics teachers in Croatia was estimated based on annual plans published on websites of the 874 elementary schools and 315 secondary schools. The list of all schools in Croatia is available on the official website of the Croatian Ministry of Science and Education [41,42]. To the principals of schools that did not have published data on the number and titles of employees in the physics teacher position in the 2022/2023 school year, we sent email requests citing the Right of Access to Information Act [43]. We repeated the request to the principals who did not respond within 10 days, and finally there were 7 (0.79%) elementary schools, and 4 (1.25%) secondary schools that did not respond, i.e., they did not provide us with the requested information.

We contacted all principals of elementary and secondary schools where physics education takes place. Principals were informed of the purpose of the survey and asked to forward our e-mail with the invitation and link to the survey to all physics teachers at their schools. Their e-mail addresses (*N* = 1200) were available to us through the official website of the Ministry of Science and Education [41,42]. The survey was open from 3 April 2023 to

15 May 2023. In the meantime, we repeated the mail to the principals to try to reach the most participants.

Of the 390 participants who answered the survey, 68% were women and 32% were men. In total, 12% of them did not complete study programs in physics and worked as non-professional substitutes for physics teachers; 78% of the participants had work experience in elementary school, 42% in high school, 19% in three-year vocational school, and 42% in four-year vocational school (Figure 1). This suggests that some physics teachers worked in several schools during the 2022/2023 school year (according to the data in the annual plans and programs of primary and secondary schools for the 2022/2023 school year) and/or in earlier school years in their careers.

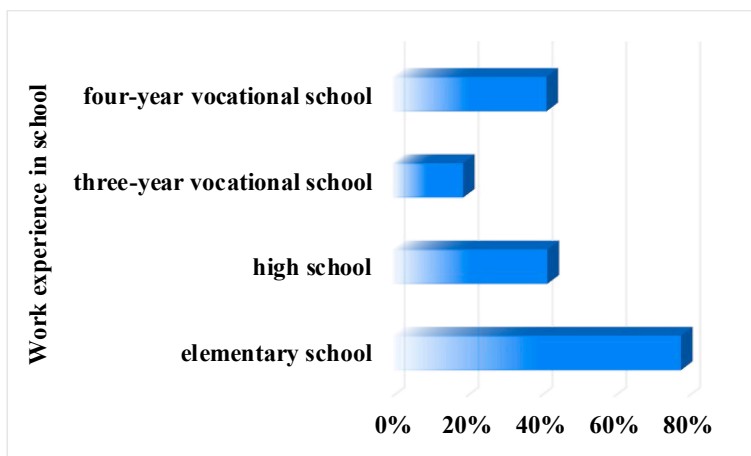

**Figure 1.** Percentage of teachers from the total number of participants in the survey with work experience in elementary school, high school, and three- and four-year vocational schools.

Teachers with an average length of service of 14.8 years participated in the survey, with the shortest length of service being 3 months and the longest being 43 years. All counties were represented, with the largest number of respondents from the City of Zagreb (18.2%), followed by Split-Dalmatia county (12.8%) and Primorje-Gorski Kotar county (9.2%) (Figure 2). The different share of participants from different counties is not a surprise if we know that the counties differ when it comes to their size and number of physics teachers in their schools.

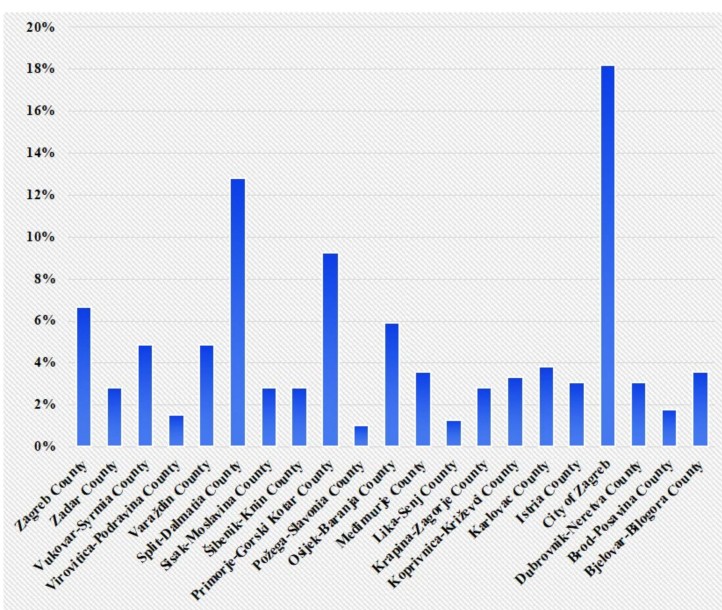

**Figure 2.** Percentage of respondents by Croatian counties.

### 2.2. Questionnaire

In the introductory section of the questionnaire, we introduced the purpose of the study to the respondents and that we were committed to using the results of the survey solely for the purposes of the study while maintaining the confidentiality and anonymity of the data collected.

The first part of the questionnaire examines the demographic characteristics of the respondents and consists of three questions on: (i) gender, (ii) schools where they have had work experience, (iii) county where they work, (iv) name of degree completed, and (v) years of work experience at the school.

The second part of the questionnaire included 40 items on the 5-point Likert scale regarding the possible causes of the shortage of physics teachers in Croatia [38]. Respondents indicated the extent to which they agreed with 40 statements (listed in Table 1 in the Results section) that reflect possible causes of the above-mentioned shortage, which can be found in the literature (see Introduction section) and were adapted to the circumstances in Croatia. In fact, we included all the causes of the shortage of STEM teachers that we found in the literature, and that can be applied to physics teachers. At the same time, we had to adapt some questions to the circumstances in Croatia. For example, there are no "district managers" in Croatia, but a similar function is performed by "school founders". We also had to add, for example, the support of the Education and Teacher Training Agency, since the Agency's activity is to carry out professional and advisory tasks in the field of elementary and secondary education. Each of the 40 statements was associated with one of the following broader categories, introduced by Mogashoa [1]: incentives, support, content of the physics course, motivation, attitudes, professional development, opportunities, and challenges. For each of the possible causes of the shortage of physics teachers in Croatia, the respondents could choose to: (1) strongly disagree, (2) disagree, (3) neither agree nor disagree, (4) agree, (5) strongly agree. Unlike Mogashoa, who used a four-category scale [3], we used a five-category scale adding the central category "neither agree nor disagree" [38]. In this way, we avoided forcing a choice in one direction that might lead respondents to have an opinion on things about which they actually have no opinion. At the end of the questionnaire, we also added an open-ended question, "Do you have any comment or additional reason for the deficit?" to give participants the opportunity to express their opinion not included in the questionnaire [38].

**Table 1.** Causes of the shortage of physics teachers in Croatia.

| The Shortage of Physics Teachers in Croatia is Due to the Following Factors: | Strongly Disagree | Disagree | Neither Agree nor Disagree | Agree | Strongly Agree |
|---|---|---|---|---|---|
| INCENTIVES | | | | | |
| 1. The salary is low. | 0.8% | 2.6% | 19.2% | 28.7% | 48.7% |
| 2. There is a lack of incentives (scholarships, favorable loans, funding for postgraduate studies, etc.). | 4.4% | 2.8% | 21.3% | 29.0% | 42.6% |
| 3. The salaries of physics teachers are unfairly equal to those of teachers of other subjects, even though the study of physics is more demanding than many other subjects. | 9.7% | 5.6% | 22.8% | 24.1% | 37.7% |
| SUPPORT | | | | | |
| 4. School administration support is insufficient. | 13.8% | 17.7% | 33.8% | 16.9% | 17.7% |
| 5. The support of the school founder is insufficient. | 6.2% | 9.5% | 29.5% | 25.1% | 29.7% |
| 6. Education and Teacher Training Agency support is insufficient. | 3.8% | 8.5% | 27.4% | 26.7% | 33.6% |
| 7. University support is insufficient. | 7.2% | 15.9% | 41.3% | 18.2% | 17.4% |
| 8. There is no clear and effective strategy of the Ministry for the employment of physics teachers. | 3.1% | 2.3% | 13.6% | 21.5% | 59.5% |
| 9. There is no clear and effective strategy of the Ministry for the retention of physics teachers in their profession. | 1.5% | 0.8% | 7.9% | 17.9% | 71.8% |
| 10. Professional association support is insufficient. | 3.8% | 7.2% | 32.3% | 29.0% | 27.7% |
| 11. There is a lack of material resources for teaching physics—cabinets, practicums, technical support, etc. | 3.3% | 7.2% | 14.9% | 24.9% | 49.7% |

**Table 1.** *Cont.*

| The Shortage of Physics Teachers in Croatia is Due to the Following Factors: | Strongly Disagree | Disagree | Neither Agree nor Disagree | Agree | Strongly Agree |
|---|---|---|---|---|---|
| **CONTENT OF THE PHYSICS COURSE** | | | | | |
| 12. Physics has a mathematical background that is too demanding. | 26.7% | 24.4% | 29.5% | 13.6% | 5.9% |
| 13. Physics is difficult to teach. | 24.9% | 19.5% | 22.3% | 21.3% | 12.1% |
| 14. Physics is more abstract than other school subjects. | 16.2% | 15.1% | 23.6% | 30.8% | 14.4% |
| 15. In textbook examples with physics content, the subjects are mostly men, so it is mostly men who choose careers in physics. | 56.2% | 17.4% | 22.3% | 3.6% | 0.5% |
| 16. University study programs for the education of future elementary school physics teachers are too demanding. | 17.2% | 16.9% | 29.5% | 23.6% | 12.8% |
| 17. University study programs for the education of future high school physics teachers are too demanding. | 17.7% | 18.5% | 29.5% | 21.8% | 12.6% |
| **MOTIVATION** | | | | | |
| 18. There are too few physicists who are popular with the public and are role models for young people. | 7.2% | 10.0% | 23.3% | 33.6% | 25.9% |
| 19. Teaching physics is boring. | 82.3% | 12.1% | 4.1% | 0.8% | 0.8% |
| 20. Physics teachers work with too little enthusiasm, so they do not inspire students to learn physics. | 36.9% | 24.4% | 26.9% | 9.5% | 2.3% |
| 21. Students' first experiences with physics classes are poor and demotivate them to continue learning physics. | 18.5% | 17.9% | 34.4% | 22.8% | 6.4% |
| **ATTITUDES** | | | | | |
| 22. The profession of physics teacher is suitable for women. | 34.1% | 6.7% | 25.1% | 5.4% | 28.7% |
| 23. The profession of physicist is suitable for men. | 41.3% | 7.7% | 26.4% | 3.3% | 21.3% |
| 24. Teachers have a bad public image, i.e., a low social status. | 3.8% | 7.2% | 15.1% | 29.0% | 44.9% |
| 25. Physics teachers have low self-confidence, so they cannot impose themselves as authorities in society. | 45.4% | 20.3% | 22.8% | 8.2% | 3.3% |
| 26. Students have a negative attitude toward physics. | 9.7% | 11.0% | 31.0% | 31.0% | 17.2% |
| **PROFESSIONAL DEVELOPMENT** | | | | | |
| 27. There are not enough university study programs for the education of future physics teachers. | 30.8% | 17.4% | 31.8% | 11.8% | 8.2% |
| 28. University study programs for the education of future physics teachers do not sufficiently develop practical competencies. | 13.6% | 13.3% | 33.1% | 26.2% | 13.8% |
| 29. Physics teachers are not offered quality in-service programs. | 10.8% | 14.1% | 34.4% | 28.2% | 12.6% |
| 30. The requirements for admission to the physics study program for teachers are too strict. | 41.3% | 20.3% | 28.7% | 5.9% | 3.8% |
| 31. There is a lack of lifelong learning programs to retrain non-physicists to become physics teachers. | 21.5% | 10.0% | 40.5% | 13.8% | 14.1% |
| **OPPORTUNITIES** | | | | | |
| 32. Physicists have more opportunities in the labor market compared to experts from other fields. | 10.3% | 10.3% | 32.1% | 28.5% | 19.0% |
| 33. Employees with knowledge of physics are increasingly needed in industry. | 4.6% | 5.6% | 26.4% | 34.9% | 28.5% |
| 34. Physics as a subject is not equally available to students in schools with different educational profiles, so not everyone has the same opportunity to experience the benefits of physics education. | 5.9% | 6.2% | 32.6% | 32.3% | 23.1% |
| 35. Many students are taught by unprofessional physics teachers who cannot ensure the quality of physics education. | 4.1% | 3.8% | 22.6% | 27.9% | 41.5% |
| 36. Physics teachers take jobs that pay better than teaching in schools. | 0.3% | 1.0% | 10.3% | 36.7% | 51.8% |
| **CHALLENGES** | | | | | |
| 37. Students are undisciplined. | 21.0% | 18.5% | 30.3% | 19.5% | 10.8% |
| 38. Students demand too much care and responsibility from the teachers. | 9.2% | 11.5% | 34.6% | 31.3% | 13.3% |
| 39. Retired teachers do not have the opportunity to work in a school. | 9.0% | 14.6% | 41.0% | 17.4% | 17.9% |
| 40. Physics teachers have an excessive workload compared to teachers of other subjects. | 4.6% | 6.4% | 22.1% | 28.7% | 38.2% |

Evidence for the validity of our questionnaire is related to the fact that the most of the 40 questionnaire statements come from a published questionnaire on *factors associated with the shortage of physical science teachers* [3]. In other words, the large majority of the questionnaire items already passed the peer-review test for their relevance when it comes to gathering information about the causes of teacher shortage. However, for the purposes of collecting additional evidence for the questionnaire's content validity in the Croatian

context, four authors of the given manuscript together with an external university professor with expertise in physics education, checked each of the 40 questionnaire statements and came to a consensus that they are relevant for measuring the respondents' opinions on the shortage of physics teachers in Croatia. They also agreed that the contents, wording, and length of items are appropriate for the sample being targeted, i.e., physics teachers from Croatia. In addition, none of the teachers in our sample claimed that any of the items were confusing or that they did not understand anything, so we can be reasonably confident that the questionnaire worked as intended.

Taking into account that we did not intend to analyze data on a composite score level, but on the individual item level, there was no need to develop psychometric scales and estimate their reliabilities.

## 3. Results

The results of the survey are presented in Table 1, which shows the percentages of teachers (N = 390) who *strongly disagreed*, *disagreed*, *neither agreed nor disagreed*, *agreed*, or *strongly agreed* with statements 1–40. The statements represent possible causes for the shortage of physics teachers in Croatia. The causes are related to incentives, support, content of the Physics course, motivation, attitudes, professional development, opportunities, and challenges for teachers, which are also the names of the corresponding categories.

When it comes to teachers' answers to the open-ended question, they often simply corroborated the causes of teacher shortage that had been already mentioned within the Likert-type items. Some of the most illustrative responses, that contained new information about the physics teacher shortage in Croatia, have been included into our Discussion section.

## 4. Discussion

The results of the survey were discussed according to the categories of the corresponding causes of the shortage of physics teachers introduced by Mogashoa [1].

### 4.1. Incentives

The majority of respondents agree or strongly agree that there is a lack of incentives in Croatia that would arouse greater interest in the physics teacher profession, which is in line with the results of earlier international studies, for example [1,18,21]. They believe that the salary is low (77.4%), that there is a lack of scholarships, favorable loans, funding for postgraduate studies, etc. (71.6%), and that the salaries of physics teachers are unfairly equal to the salaries of teachers of other subjects, even though the study of physics is more demanding than many other subjects (61.8%). Deeper insight into these issues may be gained from the teachers' answers to our open-ended question. For example, one respondent states:

> *"I plan to quit my job as a teacher because, for example, all my friends from primary and secondary school who are in trade or have started a family business are buying cars and apartments, going to the seaside and the like, while I think about how to pay the rent and registration of the car in the same month, even though I have a university degree, which is not easy."*

Respondents also suggest some new incentives, such as a salary supplement for deficit teacher professions and/or teachers who do their job well. The following comments support this view:

> *"Shortage occupations in schools should receive a salary supplement, at least 20%."*

> *"As long as someone who works responsibly, constantly improves professionally, whose students achieve remarkable results, who participates in various projects, investigates, etc., has the same or even a lower salary, e.g., due to lower seniority, compared to someone who works sloppily and carelessly, there is no way out!"*

In Table 2, the values of the average net and gross salaries paid per employee in the elementary school sector were compared with the average net and gross salaries paid per employee in the legal entities of the Republic of Croatia (average of the Republic of Croatia for the total employed population). Legal entities include employees of all forms of ownership, state government bodies, and bodies of local and regional self-government units on the territory of the Republic of Croatia. All levels of education are included (from basic education to postgraduate university doctorate). Employees in crafts and liberal professions and insured farmers are not included. Based on the given data from Table 2, we find that the average monthly salary of elementary school teachers is almost the same (even slightly higher) than the average of the Republic of Croatia, but it is relatively safe to conclude that teacher salaries are lower than the salaries of other employees with a similar level of education.

**Table 2.** Comparison of the values of average net and gross salaries per employee in primary education with average net and gross salaries per employee in legal entities of the Republic of Croatia (average of the Republic of Croatia for the total employed population), for January–April 2023 [44–48].

| Amount of the Average Monthly Salary: | January 2023 | February 2023 | March 2023 | April 2023 |
|---|---|---|---|---|
| per employee in elementary education (net) | 1139 € | 1126 € | 1154 € | 1212 € |
| per employee in elementary education (gross) | 1493 € | 1482 € | 1505 € | 1690 € |
| per employee in the legal entities of the Republic of Croatia (net) | 1094 € | 1106 € | 1130 € | 1122 € |
| per employee in the legal entities of the Republic of Croatia (gross) | 1499 € | 1522 € | 1556 € | 1547 € |

Additionally, the annual rate of change (increase) of salaries in education is one of the lowest according to the 2007 National Classification for the Economic Activities [44–46,48]. Only four activity sectors (out of 20 in total) have lower rates of salary increase compared to education. Thus, although the average teacher salaries are comparable to the average salaries in the Republic of Croatia (Table 2), their increase lags significantly behind the increase in the cost of living. For example, according to the National Bureau of Statistics, the share of food in the monthly expenses of an average Croatian household has increased from 27.2% to over 40% from the pre-pandemic and war crisis period to the present, and housing costs have increased from 17.7% to over 30%, while average salaries have increased minimally.

As we have seen in the comments above, some Croatian teachers indirectly emphasize the need to develop a quality improvement and assurance system in schools. In this context, it is useful to note that the Croatian elementary school system is planned to be subjected to external quality evaluations after an unsuccessful attempt to implement quality assurance based on the principle of self-evaluation and after the pilot project External Evaluation of Elementary and Secondary schools was conducted in the period from 2017 to 2019 by the National Center for External Evaluation of Education [49–51].

In the Digital Croatia Strategy for the period until 2032, the Government and Parliament of the Republic of Croatia refer to the level of salaries and the departure of teachers from the educational sector [52]. As one of Croatia's main weaknesses that needs to be resolved, they cite the shortage of elementary and secondary school teachers from STEM fields (especially mathematics, physics, and computer science) who leave the education sector due to low salaries in schools.

*4.2. Support*

Participants in our survey believe that they do not receive support from relevant bodies and institutions, which, according to Gold [53], may also lead to a shortage of physics teachers. The vast majority of respondents agree or strongly agree that there is no clear and effective strategy of the Ministry for the employment of physics teachers (81.0%) and also for the retention of physics teachers in their profession (89.7%), which is in line with [3]. This is supported by the comment:

*"The Ministry has systematically refused to acknowledge the existence of the problem for twenty years, which only exacerbates the problem and prevents constructive discussion of the issue."*

When reviewing the Digital Croatia Strategy until 2032 [52] and the National Development Strategy of the Republic of Croatia until 2030 [54], it was found that there are indeed no clear and effective strategic plans to retain STEM teachers in the education sector, which is consistent with the statements of the respondents. The strategies touch on the deficit of STEM in Croatia in general and refer to the problem of attracting and retaining STEM teachers in just one sentence, without presenting a structured and elaborated action plan. The only concrete measure so far has been the awarding of state scholarships in STEM for the academic year 2022/2023 under the National Recovery and Resilience Plan 2021–2026 [55], to full-time students enrolled in higher education institutions in Croatia. With this measure, the state aims to help increase the retention rate of students in STEM study programs, i.e., increase the graduation rate. However, the impact of this measure on reducing the percentage of non-professional substitutes for physics teaching in schools could possibly only become apparent after a longer period of time. There are certainly doubts as the Ministry of Science and Education's investment measures in the form of scholarships for students in STEM fields within the Recovery and Resilience Plan 2014–2020 have not had the desired effect so far. Despite the mentioned scholarships, it has been found that the total percentage of students in STEM study programs (at the undergraduate level and in the first three years of the integrated study programs) decreases by an average of approximately 1500 students per year or approximately 2.5% of students per year. Therefore, it is obvious that new measures must be taken to motivate high school students to enroll in STEM studies, to encourage graduates to work in the education system, and to retain existing physics teachers in the education system.

The majority of our study participants also agree or strongly agree that there is a lack of material resources for teaching physics, such as cabinets, practicums, technical support, etc. (74.6%), and that support from the Education and Teacher Training Agency (60.3%), professional associations (56.7%), and school funders (54.8%) is insufficient. The above results are confirmed by numerous comments, such as the following:

*"Promoting STEM is important, but it should also be accompanied by better equipment for the cabinets, as well as a necessary reduction in the physics teacher workload, because inquiry-based instruction requires more preparation, especially if we have to bring materials from another classroom or cabinet."*

*"Most physics teachers who do experiments do not have lab assistants. They do not have a person responsible for running the cabinet…"*

*"Instead of making our progress easier, we are hindered at every step! If I manage to find another job, I will definitely leave!"*

*"What we need is a central website with preparations, experiments, instructions and tips for teaching physics."*

*"If local authorities (school founders) are also unwilling to address the deficit problem, then why push into a profession that no one is interested in or promotes. Finally, there is no will on the part of physicists themselves to participate in social change. After all, knowledge is irrelevant, unprofitable, unattractive, and soon physicists will be found only in centers of excellence where some enthusiastic parents will take their children because physics will not be taught in schools anyway."*

The largest percentage of respondents neither agree nor disagree that support from the school administration (34.6%) and the university (41.3%) is insufficient, suggesting that physics teachers do not expect support from school administration and the university to impact the deficit in their profession. According to the respondents, the greatest responsibility lies with the Ministry. However, its policies to address the deficit should be based on the results of university studies such as this one. The importance of such a role for the

university was not recognized by our respondents, unlike, for example, the respondents in Mogashoa's study [3]. In comments related to support from school administration, teachers most often express dissatisfaction due to unsecured resources for professional development. For example, one of them states:

> *"We pay several hundred euros for in-service training ourselves because the school has no funding…".*

This is another example of an ineffective plan from the National Development Strategy 2030 [54]. Namely, there are plans to expand professional development opportunities for teachers, which is also a prerequisite for their promotion and salary increase, but with no clear indication of who would pay for such training.

### 4.3. Content of the Physics Course

Among the possible causes of the shortage of physics teachers in Croatia related to the content of the subject Physics, there are three statements that approximately one third of the respondents (29.5%) do not consider relevant. They neither agree nor disagree with the statements about the mathematical background of physics being too demanding and university study programs for the education of future physics teachers being too demanding, as possible causes for the shortage of physics teachers. Nevertheless, the largest percentage of respondents disagrees or strongly disagrees with statements 12 and 17 in Table 1. They believe that: (i) the mathematical background of physics is not too demanding (51.1%) to the extent that this would lead to a shortage of physics teachers; on the other hand, the majority of respondents in the study point out the importance of the role of mathematics in physics [3]. (ii) Respondents in Croatia also believe that university study programs for the education of future high school physics teachers are not too demanding (36.2%); therefore, they cannot have a negative impact on employment and retention in the physics teaching profession.

On the other hand, many respondents from our sample (36.4%) agreed or strongly agreed with the statement that university study programs for the education of future elementary school physics teachers are too demanding and are a possible cause of the physics teacher shortage. Moreover, among the comments, there is the statement that *"too hard a study program is the number one reason"* for the deficit. The largest percentage of respondents (45.2%) also agree or strongly agree that physics is more abstract than other school subjects, but disagree or strongly disagree that physics is difficult to teach (44.4%), and that in textbook examples with physics content, the subjects are mostly male, so it is mostly men who choose careers in physics (73.6%), which is consistent with the results from [3].

### 4.4. Motivation

Our respondents do not see a problem in the teachers' work when it comes to the low motivation of students to learn physics, which ultimately leads to giving up the physics teacher profession. This is supported by the fact that the largest percentage of respondents disagreed or strongly disagreed with the statement that teaching physics is boring (94.4%), and that physics teachers work with too little enthusiasm, so they do not inspire students to learn physics (61.3%), which is in line with the results in [3]. Based on the teachers' comments, it can be concluded that their long-term (15 years on average) enthusiasm is based solely on intrinsic motivation. Examples of such comments are:

> *"Teachers' salaries are too low, and all those currently teaching physics are only in the schools because they like teaching."*

> *"You mentioned everything, low salary, inferior status in society compared to physicists employed in other sectors. Insufficient extrinsic motivation… The question that arises: Is intrinsic enough to stay? And if so, until when?"*

> *"A system that survives on the enthusiasm of individuals is doomed."*

The majority of our respondents also disagree or strongly disagree with the statement that students' first experiences with physics classes are poor and demotivate them to continue learning physics (36.4%), noting that as many as 34.4% of respondents have no opinion on the impact of students' first experiences on further physics learning, which is in contrast to the results of similar studies in other countries, e.g., [1,5,51]. The majority of participants (59.5%) believe that students are not motivated to learn physics because there are too few physicists who are popular with the public and are role models for young people.

*4.5. Attitudes*

Contrary to the research findings described, for example, in [3,56], our respondents believe that attitudes related to gender inequality are not the cause of the shortage of physics teachers in Croatia. This is supported by the fact that the majority of respondents disagree or strongly disagree with the statements that the profession of a physics teacher is suitable for women (40.8%) and that the profession of a physicist is suitable for men (49.0%).

Our participants agree or strongly agree that students have a negative attitude toward physics (48.2%) and that teachers have a bad public image, i.e., a low social status (73.9%), which contributes to the shortage of physics teachers [3,56]. The following two comments support the above findings:

> *"Physics requires students to think and make connections, and most of them don't have the will to do it. . . it's hard to learn physics by heart."*

> *"The whole society, starting with politicians, badmouths us and considers us "the greatest evil in the world." Everyone thinks they have the right to tell us anything that comes to mind, including students, and no one is held accountable."*

However, our participants do not believe that low self-confidence is the reason for their inability to impose themselves as authorities in society (65.7%). This result, as well as the results related to the motivation claims, indicates that teachers believe that they cannot influence social changes on their own that would contribute to solving the problem of deficits in their profession. Therefore, they necessarily need the support of the Ministry, the Teacher Training Agency, the professional association, and school founders, which is consistent with the results related to the support claims.

Low social status as one of the causes of the shortage of professional physics teachers in Croatia is also mentioned in the analytical document "Education and skills" [57], prepared by the World Bank Group (WBG) team during the preparation of the proposal for the National Development Strategy of the Republic of Croatia for the period until 2030. Based on the statistical analysis of the data until 2019, the WBG team recommended to the Croatian government that in order to increase the attractiveness of the teaching profession and the social status of teachers, it is necessary to increase salaries, provide paid professional training, ensure attractive promotion opportunities, etc.

*4.6. Professional Development*

Many respondents disagree or strongly disagree that there are not enough university study programs for the education of future physics teachers (48.2%), and that the requirements for admission to the physics study program for teachers are too strict (61.6%), which is in contrast to the results from [3]. On the other hand, many respondents agree or strongly agree that existing university study programs for the education of future physics teachers do not sufficiently develop practical competencies (40.90%), and that physics teachers are not offered quality in-service programs (40.8%). Examples of comments supporting the previous two findings are:

> *"There is no communication between the university and the labor market. Working in a school requires, more than intellectual effort, engagement related to the upbringing of pupils. And I did not really encounter such pedagogical components during my study. And then physicists who know how to solve various mathematical problems break down on such a simple upbringing issue, which is present in school 80% of the time."*

*"When physics teachers come into schools, in-service training that focuses on physics is usually prevented. Most in-service training focuses on pedagogy and methodology, which generally applies to all subjects, not just physics."*

Many respondents (40.5%) had no opinion on the statement that there is a lack of lifelong learning programs to retrain non-physicists to become physics teachers. This is the expected result, considering that 88% of the respondents have completed study programs in physics, so information about retraining programs is not of interest to them. In contrast, the remaining 12% of respondents with completed study programs outside of physics are employed as non-professional substitutes and do not have a secure job [37]. They see retraining programs as a solution to their precarious work situation, but also as a solution to the problem of the shortage of physics teachers in Croatia, following the example in the USA [58]. For example, two participants stated:

*"As a non-professional physics teacher who already has 6 years of work experience in elementary school, I think non-professional teachers do not have the opportunity to teach students in the long run. At least a lifelong learning program to retrain non-physicists to become physics teachers is needed."*

*"It would be good if there was a part-time study program in physics…"*

### 4.7. Opportunities

The majority of teachers who participated in the survey agree or strongly agree with all statements related to lack of opportunities as possible causes of the shortage of physics teachers in Croatia: (i) physicists have more opportunities on the labor market compared to experts from other fields (47.5%), (ii) employees with knowledge of physics are increasingly needed in industry (63.4%), (iii) physics as a subject is not equally available to students in schools with different educational profiles, so not everyone has the same opportunity to experience the benefits of physics education (55.4%), (iv) many students are taught by unprofessional physics teachers who cannot ensure the quality of physics education (69.4%), and (v) physics teachers take jobs that pay better than teaching in schools (88.5%). The agreement with the first three statements is consistent with the results from [3], and the remaining two statements can be supported by comments:

*"There are many unprofessional substitutes who reduce physics education to reproducing published preparations, which results in physics education being reduced to solving numerical problems without conceptual understanding."*

*"Salaries are not comparable to salaries in other sectors where we are offered jobs."*

### 4.8. Challenges

Regarding challenges, the largest number of respondents agree or strongly agree with the statements that physics teachers have an excessive workload compared to teachers of other subjects (66.9%), which is in line with earlier research [3,6], and that students demand too much care and responsibility from the teachers (44.6%). In the comments, one teacher suggests a specific measure to relieve workload:

*"In elementary school, the workload is 24 h per week for direct instruction. Many teachers accumulate this in 3–4 schools. In addition, there are other duties such as handling equipment, which are often passed from school to school. Designing experiments and maintaining equipment should reduce the hourly rate for direct instruction by at least 2 h per week."*

There are also examples of comments describing the increasing demands of working with children, such as the following:

*"Working with children is becoming more challenging due to discipline problems, an increasing number of students needing individual help, and increasing problems with student attention…"*

Respondents also disagree or strongly disagree with the statement that students are undisciplined (39.5%), unlike the majority of respondents from [3]. Indeed, undisciplined students can disrupt physics classes and demotivate other students to learn physics [3], as well as teachers to teach physics [6]. Of interest is the comment by a teacher who sees and explains the problem as not with the students but with the parents:

> *"There is a great dislike for undisciplined parents (not students!) who think their children must have an excellent grade. It's sad that adults do not understand that regardless of (lack of) knowledge, we all play a role in shaping the world."*

The largest percentage of Croatian respondents neither agree nor disagree (41.0%) with the statement that retired teachers do not have the opportunity to work in a school, suggesting that they are not sure about this factor as a possible cause of the physics teacher shortage.

*4.9. Research Limitations*

A limitation of this study is related to the fact that no piloting of the questionnaire has been performed and only little validity evidence has been collected to date. However, taking into account that the large majority of questionnaire statements comes from a published instrument that already passed a peer review test, as well as the fact that a panel of experts additionally checked the Croatian version of the instrument, we can be reasonably confident that the questionnaire worked as intended. In fact, none of the 390 teachers reported issues in understanding some of the statements.

## 5. Conclusions and Future Research

The shortage of physics teachers is a global and persistent problem. Therefore, in order to develop effective educational policies, it is necessary to know the causes of the physics teacher shortage as well as the most prevalent reasons for choosing physics teaching as a profession. A review of the literature revealed the following factors that positively affect the employment and retention of physics teachers: positive experiences with mentors, a family tradition of teaching, positive perceptions of teachers' work schedules, and positive perceptions of working with children. On the other hand, there are numerous examples of the causes of the physics teacher shortage, such as low salaries, lack of incentives, negative perceptions of working with children, low self-esteem, demographic changes, perceptions of low social status of the teacher profession, high workloads, poor working conditions, lack of or poor quality of educational programs for physics teacher education, impeded professional development, etc.

Taking into account that reasons for the physics teacher shortage may strongly vary for the contexts of different countries, we decided to conduct a descriptive cross-sectional study aimed at identifying Croatia's physics teachers' opinions on the shortage of physics teachers in Croatia.

By analyzing the answers obtained from a representative sample of Croatia's physics teachers, we could come to the following conclusions about the main causes of the physics teacher shortage in Croatia: (i) there is no clear and effective strategy of the Ministry for the employment and retention of physics teachers in their profession, (ii) physics teachers take jobs that pay better than teaching in schools, (iii) the salary is low, (iv) there is a lack of material resources for teaching physics and incentives, (v) teachers have a bad public image, and (vi) many students are taught by unprofessional physics teachers who cannot ensure the quality of physics education.

We are aware that some of the items can be considered transversal to all teaching areas, not only physics, but we have not excluded them as such. In this study, as in similar ones, it has been shown that these transversal problems are indeed problems of physics teachers. Moreover, they are among the major causes of the shortage that emerge from the analysis of responses obtained exclusively from physics teachers.

All the main causes as well as the majority of all our results are consistent with the results of previous international studies. However, some of our findings contrast with the

results of similar studies. In terms of support, our respondents do not believe that students' initial experiences with physics classes have an impact on further physics learning. In terms of motivation, they do not believe that universities play an important role in addressing the shortage. Our respondents also believe that attitudes related to gender inequality are not the cause of the physics teacher shortage. Regarding professional development, they believe that there are enough university study programs for the education of future physics teachers and that the requirements for admission to the physics study program for teachers are not too strict. Related to challenges, they do not think that students are undisciplined.

Scientific studies that show concrete scientific results on the causes of the shortage are an indispensable mechanism for initiating the relevant institutions to jointly design future actions to combat the aforementioned problem. Therefore, the academic implications of the present work are very significant in the way that this study, together with our previous study, represents the only scientific analysis of the shortage of physics teachers in Croatia and the possible causes of the shortage. The importance of this study lies in the fact that it will most likely trigger all relevant institutions, such as governments, universities, and schools, to deal with the results and to jointly develop concrete steps to solve the problem. Therefore, in cooperation with all relevant stakeholders, previous political measures on the "trial and error" principle can be converted to successful policies.

The limitations of this paper manifest in the recognition that solving the problem of the shortage of physics teachers (or STEM) depends not only on us, the university stakeholders, but also on the future commitment of other stakeholders and on the fact that concrete actions and changes will require a longer period of time. We expect that based on specific scientific results from this paper (no clear and effective strategy of the Ministry, low salary, bad public image, a lack of material resources, many students are taught by unprofessional physics teachers, unpaid professional trainings) and the results of our next study, the Ministry should consider introducing specific actions in its strategic plans or even create a specific strategic plan addressing only the teacher shortage in STEM (similar to the plan addressing the shortage of IT teachers in the Digital Croatia Strategy). We provide possible solutions regarding the educational causes of the shortage of physics teachers in Croatia: (i) development of occupational standards and qualifications with the aim of improving the quality of study programs and their connection with the needs of the labor market, (ii) development of educational physics modules as parts of university undergraduate study programs, possibly as short-time courses with subjects in physics, education, and teaching practice, which could positively affect the recruitment and retention of employees in the physics teacher positions, (iii) organizing of lifelong learning with the aim of improving knowledge in physics and developing pedagogical skills of non-professional teachers, (iv) organizing of paid professional development activities and training, and (v) increase in salaries. University stakeholders can affect solutions (i)–(iii), but unfortunately cannot affect solutions (iv) and (v), for which the initiative of the government and Ministry is needed.

As mentioned in our earlier work, the problem of the shortage of physics teachers in Croatian schools directly affects the negative attitude of students towards physics. This is also reflected in students' grades in the State Matura in physics, which are regularly below average, or in the below average results of Croatian students in the Program for International Student Assessment (PISA), which assesses students' knowledge and skills in science. Devastatingly, recent State Matura high school graduation results indicate that Croatian high school students performed the worst in physics (46.4 percent of students failed the physics exam) [59].

In the next phase we plan to develop and propose a concrete action plan that is based on the results of this study and, at the same time, is consistent with existing strategies and analytical documents in the Republic of Croatia. There is also the possibility of joining forces with colleagues from neighboring countries (Serbia and Bosnia and Herzegovina) to extend the studies to a wider region, as there is a shortage of physics teachers in these areas as well. Our recommendation to researchers in other countries with the same problem

is to try to influence the relevant institutions to make changes based on scientific results if the relevant institutions themselves do not consciously address the problem and do not make an effort to solve it. A shining example of solving the problem of teacher shortage in general, focusing on STEM (with low attractiveness of the profession and low wages as causes), is the action of the Estonian and Latvian governments, which in recent years enabled higher teacher salaries than the average salaries in these countries, believing that salaries are one of the most important means to convince more talented people to enter the teaching profession [60,61].

**Author Contributions:** Conceptualization, N.E.; methodology, N.E., V.M. and L.N.; validation, N.E., V.M., L.N. and I.P.B.; formal analysis, N.E. and L.J.; investigation, N.E., L.J. and P.N.; resources, N.E. and I.P.B.; data curation, N.E. and L.J.; writing—original draft preparation, N.E.; writing—review and editing, N.E., I.P.B., L.J., V.M. and L.N.; visualization, N.E.; supervision, N.E.; funding acquisition, N.E. and I.P.B. All authors have read and agreed to the published version of the manuscript.

**Funding:** This research received no external funding. The APC was funded by Faculty of Physics—University of Rijeka.

**Institutional Review Board Statement:** The study was approved by the Ethics Committee of Faculty of Physics—University of Rijeka (KLASA:035-01/23-01/39, URBROJ:2170-137-003-01-23-1, 19 June 2023).

**Informed Consent Statement:** All respondents were informed about the nature of our study and they voluntarily participated in the study. The participants were assured that the principles of confidentiality and anonymity would be adhered to in the study.

**Data Availability Statement:** The data presented in this study are available on request from the corresponding author. The data are not publicly available due to ethical restrictions.

**Acknowledgments:** The authors would like to thank all the participants who took part in this study.

**Conflicts of Interest:** The authors declare no conflict of interest.

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
