# Peer review of "Causes of the Shortage of Physics Teachers in Croatia"

_education, doi:10.3390/educsci13080788_

Round 1

Reviewer 1 Report

Reviewer’s suggestions and comments on the Manuscript entitled:

Causes of the shortage of physics teachers in Croatia

Manuscript ID: education-2511120

This excellent study of colleagues from Croatia opens a rather unpleasant question regarding the shortage of physics teachers.  This issue is global, and soon it can have drastic consequences. Therefore this manuscript has high citation potency. I strongly recommend that the Editorial Office accept this manuscript after minor revision.

Suggestions:

Figure 2. Percentage of respondents by Croatian counties. Some colors have to be used, otherwise, I don’t see intent. The figure is too pale.  

Line 160: ˮforty 5-point Likert-type itemsˮ is it 45 or?

I don’t see a reason-why for Figure 3, this figure is too complicated to be observed, however, if authors want to keep it is fine by me. Maybe this figure can be broader.

Reviewer 2 Report

Dear authors,

The study presented in this manuscript has a pertinent focus with regard to the concerns with the shortage of (physics) teachers and its causes.

The document is globally well organized and written in a clear and comprehensive way.

The following comments and suggestions intend to help improve the quality of the paper and are organized concerning its’ different sections:

- To be complete, the abstract should identify the method used.

- The problem should be clearly delimited in the introduction and written in the same way as in the abstract.

- Introduction:

            I recommend that the authors’ add more recent references to make some assumptions more robust. The used majority of the references are not very recent.

- Methodology:

The authors should identify the paradigm/methodology that underlies the study, not only the method.

More information should be included about the procedures that resulted in the participation of 390 respondents (how were they selected, how were they contacted, how did you try to ensure the most participants, …).

Despite of stating that the questionnaire was analized by a team of experts, how was it validated? Was the questionnaire piloted?

How did the authors choose the questions for the second part of the questionnaire? Which criteria did they use? 

- Discussion

            Some of the items can be considered transversal to all teaching areas, not only physics. How do the authors guarantee the specificity of the area, in order to explain the shortage in physics teachers?

- Conclusions:

            The discussion in the conclusions must be crossed with literature.

            It would be important to reflect about the differences drawn from this study in comparison to the already known data.

No comments.

Reviewer 3 Report

The paper is well written with interesting applications. The methodology used is adequate with concise and logic conclusions. There is clear scientific contribution of paper.

The structure of the conslusions should be improved. Please add academic implications, limitations of the paper and future studies and recommendations.
